# Registration Quality Evaluation Metric with Self-Supervised Siamese Networks

**Tanvi Kulkarni**[1,2]                                                      EE20S046@SMAIL.IITM.AC.IN
[1] *Department of Electrical Engineering, Indian Institute of Technology Madras (IITM), India*
[2] *Healthcare Technology Innovation Centre, IITM, India*

**Sriprabha Ramanarayanan**[1,2]                                      SRIPRABHA.R@HTIC.IITM.AC.IN
**Keerthi Ram**[2]                                                          KEERTHI@HTIC.IITM.AC.IN
**Mohanasankar Sivaprakasam**[1,2]                                      MOHAN@EE.IITM.AC.IN

**Editors:** Accepted for publication at MIDL 2024

## Abstract

Registration is one of the most preliminary steps in many medical imaging downstream tasks. The registration quality determines the quality of the downstream task. Traditionally, registration quality evaluation is performed with pixel-wise metrics like Mean Squared Error (MSE) and Structural Similarity Index (SSIM). These pixel-wise measures are sometimes susceptible to local minima, providing sub-optimal and inconsistent quality evaluation. Moreover, it might be essential to incorporate semantic features crucial for human visual perception of the registration quality. Towards this end, we propose a data-driven approach to learn the semantic similarity between the registered and target images to ensure a perceptual and consistent evaluation of the registration quality. In this work, we train a Siamese network to classify registered and synthetically misaligned pairs of images. We leverage the latent Siamese encodings to formulate a semantic registration evaluation metric, SiamRegQC. We analyze SiamRegQC from different perspectives: robustness to local minima or smoothness of evaluation metric, sensitivity to smaller misalignment errors, consistency with visual inspection, and statistically significant evaluation of registration algorithms with a p-value $< 0.05$. We demonstrate the effectiveness of SiamRegQC on two downstream tasks; (i) Rigid registration of 2D histological serial sections, where evaluating sub-pixel misalignment errors is critical for accurate 3D volume reconstruction. SiamRegQC provides a more realistic quality evaluation sensitive to smaller errors and consistent with visual inspection illustrated with more perceptual semantic feature maps rather than pixel-wise MSE maps. (ii) Unsupervised multimodal non-rigid registration, where the registration framework trained with SiamRegQC as a loss function exhibits a maximum average SSIM value of 0.825 over previously proposed deep similarity metrics.

**Keywords:** Image registration, Evaluation metric, Cosine similarity, Siamese network, Semantic representation.

## 1. Introduction

Registration is the task of aligning a source image to match the physical coordinates of a target image. In medical image analysis, registration is used for many downstream tasks, such as atlas-based segmentation (Balakrishnan et al., 2019), (Kulkarni et al., 2023), (Aqil et al., 2023) and reconstruction of 3D volumes of organs/tissues by successively registering their 2D histological serial section images (Kajihara et al., 2017). The registration quality

directly impacts the effectiveness of the downstream tasks, where even small errors are significant. For instance, in 3D histological reconstruction (Lobachev et al., 2021), the misalignment error at every serial section registration accumulates from the middle section to the ends of the volume, resulting in an irregularly reconstructed 3D volume. Leveraging more accurate quality metrics for registration optimization can result in more convergent performances (Simonovsky et al., 2016), (Czolbe et al., 2023). Traditional metrics like Mean Squared Error (MSE), Structural Similarity Index (SSIM) (Wang et al., 2004), Jaccard overlap measures (Kartasalo et al., 2018), (Xu et al., 2016) and MIND (Heinrich et al., 2012) are commonly used to evaluate the registration quality. Furthermore, traditional metrics rely on basic pixel-wise calculations and can be susceptible to local minima (Zhang et al., 2018). Including semantic features to capture the nuances essential for human visual perception could ensure a consistent and perceptually accurate assessment of the registration quality (Wang and Bovik, 2009).

With the advent of Machine Learning (ML) for medical applications, several supervised ML algorithms are proposed that automatically classify affine registered and misaligned pairs of images (Sokooti et al., 2019), (Tummala et al., 2021). However, these algorithms are supervised using traditional metrics and do not offer quality assessment beyond the binary classification of misaligned pairs. Recently, deep learning (DL) methods have been proposed using semantic representations and intermediate network layers as perceptual metrics for image quality assessment (Gao et al., 2017), (Zhang et al., 2018). Such perceptual metrics are shown to provide a more reliable quality assessment than signal-to-noise ratio and SSIM for imaging applications like object detection and denoising. Previously, two-channel Convolutional Neural Networks (CNNs) have been proposed as deep similarity metrics for multimodal registration (Cheng et al., 2018), (Simonovsky et al., 2016). Interestingly, DeepSim (Czolbe et al., 2023) introduced semantic features derived from unsupervised autoencoders as similarity measures to optimize DL-based registration methods. Unlike the DeepSim method, we use Siamese networks to obtain the semantic representations of the input images efficiently (Refer Appendix A).

Siamese networks have been applied for image registration (Chen et al., 2021), (Neumann et al., 2020), (Tang et al., 2022) due to their efficiency in training a pair of input samples (multiple inputs, in general) using the same network parameters. Because of their dual encoder architecture with identical parameters, Siamese networks can transform an input pair of registered and target images into the same latent feature space (Bromley et al., 1993). The cosine similarity function is chosen to formulate our proposed registration evaluation metric due to its desirable property of evaluating similarities irrespective of the dimensionality of features and having a restricted range of values always lying between -1 and +1 as shown in (Nguyen and Bai, 2010). Additionally, to learn distinctive representations, we utilize the cosine similarity function as a contrastive loss (Chen et al., 2020) that encourages or discourages the similarities between registered or misaligned input image pairs (Refer Appendix B). To summarize our contributions, we propose SiamRegQC, a data-driven deep learning-based quality evaluation metric for image registration. The proposed metric is agnostic to the registration method and uses semantic representations of the registered and target images learned from a Siamese network. We assess the efficacy of the proposed evaluation metric, SiamRegQC, in the following aspects:

1. Robustness to local minima - SiamRegQC shows smoother surfaces than MSE and SSIM, as seen from the metric surface plot analysis over the rigid misalignment space.

2. Sensitivity to smaller misalignment errors - From the local variance measures of the surface plot analysis, SiamRegQC exhibits the best sensitivity to unit changes in the misalignment space with a value of 1.9e-3 while MSE was found to be insensitive with a sensitivity value as low as 3.5e-5.

3. Consistency with visual inspection - The latent Siamese encodings offer the advantage of a better perceptual understanding of the registration quality than pixel-wise MSE maps.

4. Application to the downstream section-wise 2D rigid registration task for Nissl-stained mouse brain volume reconstruction. Here, we use SiamRegQC as a registration quality evaluation metric for benchmarking the performance of three different registration algorithms, where SiamRegQC critically evaluates the algorithms due to its enhanced sensitivity.

5. We leverage SiamRegQC as a similarity metric to drive VoxelMorph architecture-based unsupervised non-rigid deformable registration framework for unimodal and multimodal data. A maximum average SSIM value of 0.967 (0.825 for multimodal) was observed in the registration outputs compared to previously proposed methods.

## 2. Methodology

This section introduces the overall network architecture for formulating the proposed registration evaluation metric, SiamRegQC, followed by a description of the dataset and implementation details for evaluating SiamRegQC.

### 2.1. Network Architecture

The network architecture for SiamRegQC consists of a Siamese network, as shown in Figure 1. The Siamese network is divided into two functional parts: an encoder, $\phi$, for deep feature extraction of the input images, and a fully connected network, $Clf$, for the classification task. The encoder, $\phi$, consists of 4 convolutional blocks of 128, 64, 32, and 16 channels each. Each layer has a convolutional kernel of size 3, stride 1, and ReLU nonlinear activation followed by a MaxPool layer of stride 2. The classifier, $Clf$, consists of two hidden layers, with 1024 and 256 neurons with ReLU activation (Refer Appendix B and Figure 2 for more details).

### 2.2. Loss functions and Training

SiamRegQC is trained with a two-step training procedure as shown in Figure 2. (i) The encoder, $\phi$ of SiamRegQC, is trained with an autoencoding task. Here, the MSE loss between a single input image and its decoded output is minimized. (ii) The entire network (encoder, $\phi$ + classifier, $Clf$) is trained end-to-end on a classification task based on the categorical labels (aligned- class 0 or misaligned- class 1) assigned to a pair of target and moving images as the input. We use a CrossEntropy loss, $L_{ce\_loss}$, and a modified cosine similarity-based contrastive loss (Chen et al., 2020), $L_{con\_loss}$, to drive the training. Hence, the combined loss, $L_{total}$, for training our Siamese network is formulated as:

$$L_{total} = L_{ce\_loss}(pred, label) + L_{con\_loss}(\phi(img), \phi(ref), label) \tag{1}$$

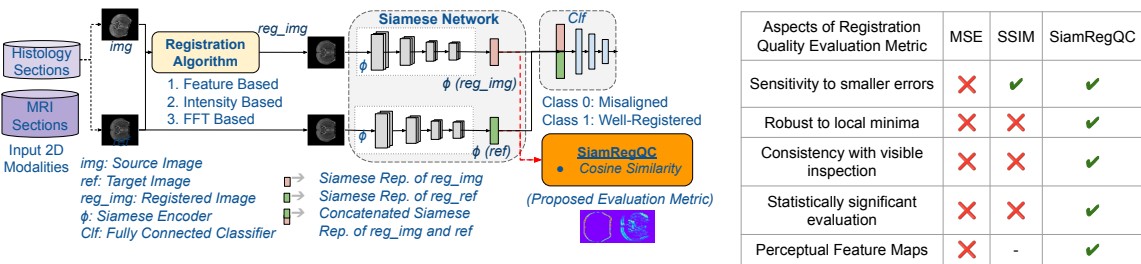

Figure 1: Graphical Representation of semantic registration evaluation metric, SiamRegQC, and its formulation. Table depicting the different desirable aspects of SiamRegQC.

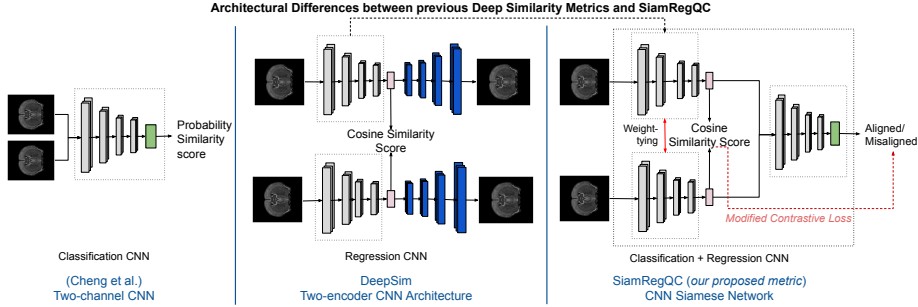

Figure 2: SiamRegQC allows for learning deep semantic features with encoders pre-trained from an autoencoding task to minimize the MSE loss between input and decoded images. In a second training step, SiamRegQC is supervised with a binary misalignment classification task to classify original and synthetically misaligned images.

where, $L_{con\_loss}$ is formulated as:

$$L_{con\_loss}\left(\phi(ref),\phi(img),label\right)=label\cdot(1-Cos\_Sim(\phi(ref),\phi(img)))^2+ \\ (1-label)\cdot Cos\_Sim((\phi(ref),\phi(img)))^2 \tag{2}$$

where, $pred$ is the predicted class, and $label$ is the true class for the input pair of images $img$ and $ref$.

## 2.3. Registration Quality Evaluation Metrics

We leverage the cosine similarity measure between the Siamese encodings of registered and target images to formulate SiamRegQC. The quality of the registration between a registered image, $img$, and a target image, $ref$, is formulated using the following equations:

$$Cos\_Sim(\phi(ref),\phi(img))=\frac{\phi(ref)\cdot\phi(img)}{||\phi(ref)||_2\cdot||\phi(img)||_2} \tag{3}$$

$$\text{SiamRegQC}(ref,img)=1-Cos\_Sim(\phi(ref),\phi(img)) \tag{4}$$

A lower value of SiamRegQC closer to 0 suggests that the input pair of images is well-registered, while a value closer to 2 (the cosine similarity function can take a minimum value of -1)suggests that the images are misaligned.

## 2.4. Dataset and Implementation Details

To evaluate registration quality in the context of a downstream task, we use a 0.05mm downsampled version of the high-resolution Nissl-stained histological adult mouse brain data with 200 coronal sections, distributed by the Allen Brain Institute (Price, 2008). We use 3000 coronal sections of adult human brain MRI volumes from the IXI dataset[1], each with a pixel resolution of 1mm to validate our method on a larger dataset. All volumes are skull-stripped, bias-corrected, and intensity normalized as described by (Chen et al., 2022). All the volumes are zero-padded and center-cropped, with each 2D section of size $256 \times 256$.

We use random rotations and translations to generate synthetic rigid misaligned images, while pixel-wise deformed images are generated using random smooth Gaussian flow fields (Refer Figure 10) All the network training experiments are run on Nvidia GeForce GTX 1660 Ti GPU with 14GB RAM for 5 epochs, in a 5-fold cross-validation method with an Adam optimizer (Kingma and Ba, 2014) of initial learning rate 0.001.

## 3. Experiments and Results

In this section, we demonstrate the effectiveness of SiamRegQC in two steps: (i) Explore the desirable aspects of SiamRegQC - robustness to local minima, sensitivity to misalignment errors, and consistency with visual inspection, as mentioned earlier. (ii) Application of SiamRegQC as a similarity measure to drive unsupervised non-rigid deformable registration on multimodal data.

## 3.1. Desirable Aspects of SiamRegQC for critical registration evaluation

In this section, we discuss the desirable aspects of SiamRegQC that lead to improved registration optimization when used as a deep-similarity metric for VoxelMorph registration, as discussed in Section 3.2.

**Robustness to local minima-** We study the variation of metric values for a maximum translation error of 0.4 mm and rotation error of 4 degrees, as shown in Figure 3. Ideally, the metric values on the surface plot should vary smoothly to avoid inconsistent evaluation. We find that MSE and SSIM are not immune to local minima, most visibly seen around no translation error and minimum rotation error. Meanwhile, all the variants of SiamRegQC show smoothly varying metric values in the rigid misalignment space. $\text{SiamRegQC}_{np}$, $\text{SiamRegQC}_{nl}$ and $\text{SiamRegQC}_{np\_nl}$ represent ablated versions of pre-training the encoder, $\phi$ and using $L_{con\_loss}$ while training SiamRegQC. Here, "$np$", "$nl$" indicates "no-pre-training", and "no-contrastive-loss", respectively. Therefore, this sensitivity analysis also stands as an ablative study to show the importance of pretraining and the addition of contrastive loss.

**Sensitivity of Evaluation Metric and Perceptual Visualization of Misalignment Errors-** From Figure 3, we calculate the mean local variance of the evaluation metric over a unit area in space to grossly represent the sensitivity of each metric, $\delta$, to every degree change of rotation error and 0.1 mm of translation error. SiamRegQC exhibits the maximum sensitivity with $\delta = 0.0019$, closely followed by SSIM with $\delta = 0.0015$. Figure 4 shows that the difference between SiamRegQC feature maps is visually more intuitive of the misalignment error than MSE difference maps. Notice that SiamRegQC has the highest

---

1. https://brain-development.org/ixi-dataset/

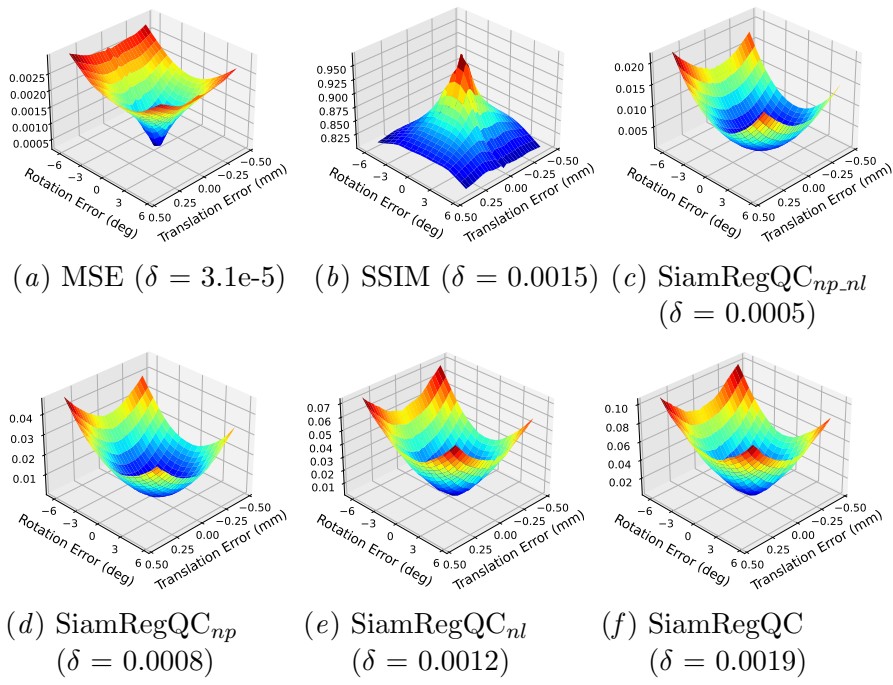

(a) MSE ($\delta = 3.1\text{e-}5$)  (b) SSIM ($\delta = 0.0015$)  (c) SiamRegQC$_{np\_nl}$
($\delta = 0.0005$)

(d) SiamRegQC$_{np}$
($\delta = 0.0008$)

(e) SiamRegQC$_{nl}$
($\delta = 0.0012$)

(f) SiamRegQC
($\delta = 0.0019$)

Figure 3: Surface Plot Analysis of registration quality evaluation metrics over the rigid misalignment space. While MSE and SSIM show the presence of local minima distinctly, SiamRegQC exhibits a fairly smoother surface. $\delta$ refers to the mean sensitivity of the metric to a change of 0.1 mm, 1° translation, and rotation errors, respectively. SiamRegQC shows maximum sensitivity to misalignment errors, $\delta$.

metric difference for a translation error difference of 0.25 mm for Figure 4a, further illustrating its high sensitivity aspect. The consistency of SiamRegQC with visual inspection and application to rigid registration algorithms are covered in Appendix C.

### 3.2. SiamRegQC as a deep similarity metric for unsupervised multimodal non-rigid registration

In this section, we use SiamRegQC as a deep-similarity-based loss function to drive unsupervised non-rigid deformable VoxelMorph registration trained on pairs of intra-subject multi-contrast T1 and T2 MRI images from the IXI dataset to test the effectiveness of our model on multimodal data, as shown in Figure 5. Figure 10 in Appendix E shows the process of generating synthetic deformable transformations for training SiamRegQC. Table 1 and Figure 6 show that SiamRegQC performs competitively better than other deep similarity metrics (Cheng et al., 2018), (Czolbe et al., 2023) and traditional multimodal metrics like Normalized Cross Correlation (NCC) and MIND (Heinrich and Hansen, 2022). Figure 7 shows corresponding qualitative examples of multimodal registration. Further exploration into other multimodal datasets and varied registration frameworks seems to be an interesting topic for future lines of work. A similar unimodal registration for IXI T1-MRI

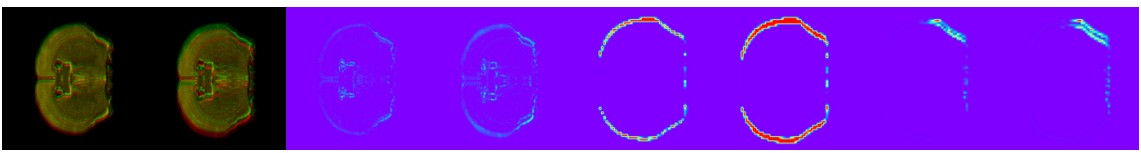

(a) NISSL Dataset; Translation Error: 0.25 mm, 0.5 mm; MSE: 0.002, 0.003; SSIM: 0.659, 0.631; SiamRegQC: 0.052, 0.132.

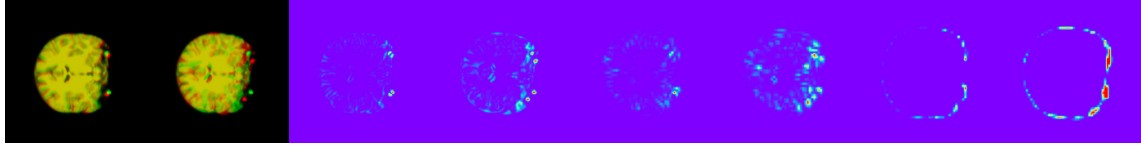

(b) MRI Dataset; Rotation Error: 4°, 14°; MSE: 0.002, 0.005; SSIM: 0.843, 0.774; SiamRegQC: 0.011, 0.090.

Figure 4: Examples showing quality evaluation maps for overlapping registered and target images. Columns 1, 2: Green is the target image, $ref$; red is the registered image to be evaluated, $img$, and yellow represents the overlapping regions between $ref$ and $img$.; Columns 3, 4: Pixel-wise MSE map, $||ref - img||_2$; Columns 5, 6 and Columns 7, 8: Channels 2 and 11 of Siamese network encoded feature activation maps,$||\phi(ref) - \phi(img)||_2$, that delineate the misalignment errors at the image boundaries.

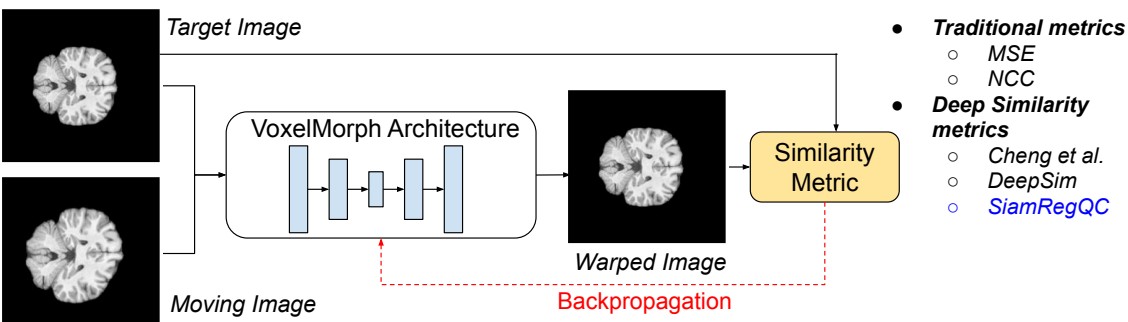

Figure 5: Incorporating SiamRegQC as loss function in unsupervised image registration framework and comparison with other traditional and deep similarity metrics.

| Evaluation Metric | Deformed Image (Before Registration) | VXM$_{ncc}$ | VXM$_{MIND}$ | VXM$_{Cheng}$ | VXM$_{DeepSim}$ | VXM$_{SiamRegQC}$ | ANTsPy |
|---|---|---|---|---|---|---|---|
| MSE | $0.022 \pm 0.021$ | $0.012 \pm 0.006$ | $0.011 \pm 0.005$ | $0.012 \pm 0.005$ | $0.012 \pm 0.005$ | $0.010 \pm 0.004$ | $0.011 \pm 0.005$ |
| NCC | $0.65 \pm 0.11$ | $0.821 \pm 0.033$ | $0.815 \pm 0.033$ | $0.837 \pm 0.005$ | $0.814 \pm 0.032$ | $0.825 \pm 0.027$ | $0.902 \pm 0.022$ |
| SSIM | $0.699 \pm 0.134$ | $0.817 \pm 0.067$ | $0.822 \pm 0.063$ | $0.834 \pm 0.059$ | $0.828 \pm 0.062$ | $0.845 \pm 0.056$ | $0.865 \pm 0.053$ |

Table 1: Quantitative Evaluation of SiamRegQC before and after registration with other traditional and deep similarity metrics for MRI T1 to T2 multimodal data. Green highlights the best evaluation metric performance, and blue highlights second best performance.

data is detailed in Appendix E, which shows that SiamRegQC performs better than other similarity metrics.

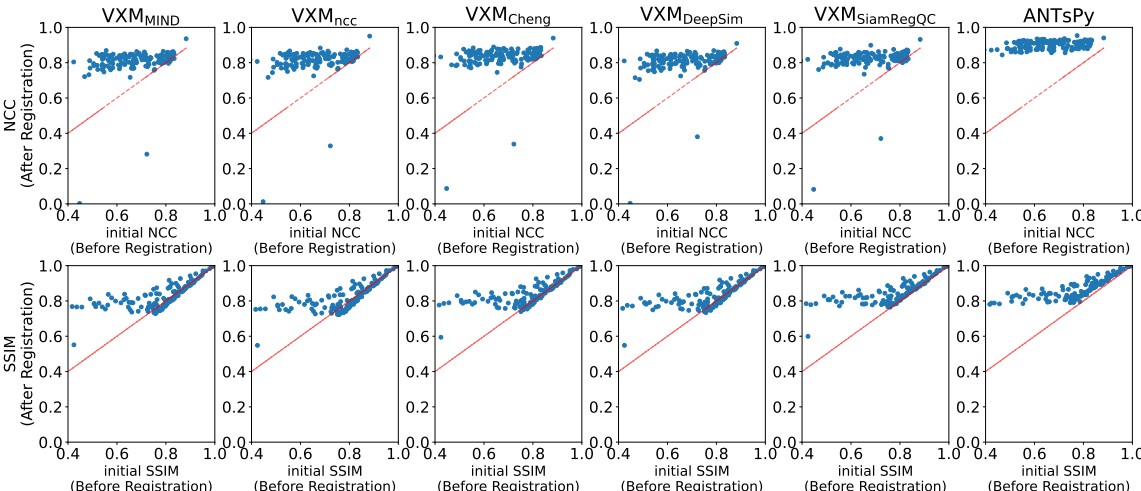

Figure 6: Improvement in NCC and SSIM scores of SiamRegQC with different similarity loss functions for MRI T1 to T2 multimodal images. SiamRegQC shows competitive improvements compared to other deep similarity-based loss functions and is closest to the reference ANTsPy performance. All data points above the dashed line suggest improvement in the registration performance.

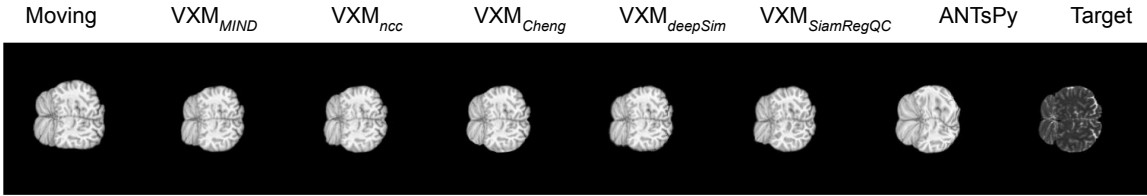

Figure 7: Registered images from different VoxelMorph networks supervised with various traditional and deep-similarity-based loss functions for MRI T1 to T2 multimodal images.

## 4. Conclusion and Future Work

In this work, we take a first step towards utilizing Siamese network-encoded representations for registration quality evaluation. We analyze our results from different perspectives. Our proposed data-driven, deep learning-based evaluation metric, SiamRegQC, is less affected by local minima and offers well-delineated registration quality visualization maps closer to human perception than pixel-wise MSE maps. SiamRegQC shows increased sensitivity to even smaller misalignment errors while maintaining consistency of values for visibly well-registered images. SiamRegQC allows for evaluating and benchmarking registration methods with statistical significance. Finally, SiamRegQC exhibits superior unsupervised deformable registration performance compared to previously proposed deep similarity metrics for unimodal and multimodal data. From a broader perspective, our paper opens up interesting directions to formulate evaluation strategies using data-driven representation learning beyond medical image registration.

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

## Appendix A. Previous Related Works

In the recent past, learning a similarity metric with supervised CNNs for optimizing a registration algorithm has been studied in many works. A CNN architecture with two-channel input has been proposed for the classification of aligned and misaligned multi-modal images (Cheng et al., 2018). They leveraged the predicted class as a probabilistic similarity score. Further improvement of the novel similarity metric has been explored by performing the misalignment classification on smaller patches of the image for better localization (Simonovsky et al., 2016). Furthermore, the authors of (Simonovsky et al., 2016) demonstrated the advantages of their proposed similarity metric by actually using it to drive a continuous optimization registration framework. DeepSim, another data-driven similarity metric, has recently been proposed for registration (Czolbe et al., 2023). DeepSim uses autoencoders trained on an unsupervised autoencoding task guided with an MSE Loss for deep feature extraction of the target and moving images. While the earlier deep similarity metrics proposed by (Cheng et al., 2018) and (Simonovsky et al., 2016) benefit from training their two-channel CNNs in the context of registration misalignment with a binary classification task, DeepSim (Czolbe et al., 2023) might have the advantage of learning more effective semantic features from a more complex autoencoding task. In this work, we aim to combine these advantages and propose an enhanced deep similarity metric using Siamese network encoders for learning complex semantic features from a misalignment classification task. Refer to Figure 2 for a graphical illustration of the architecture differences of SiamRegQC from the previously proposed methods. Another key difference between SiamRegQC and the previous metrics is that SiamRegQC is trained in two steps. (i) The encoder, $\phi$ of SiamRegQC, is trained with an unsupervised autoencoding task on a single input MRI image. (ii) The entire network (pre-trained encoder + classifier, $Clf$) is trained end-to-end on a classification task based on the categorical labels (aligned or misaligned) assigned to a pair of target and moving images as the input.

## Appendix B. Rationale for choosing the architecture for SiamRegQC

In this section, we explain the rationale for using a Siamese network architecture as the backbone for semantic feature extraction of the target and moving images. We further discuss the simplicity of using a classification task instead of a regression task at the final stage of the architecture, as shown in Figure 2.

### B.1. Rationale for using Siamese Networks

The dual-encoder architecture of SiamRegQC provides the ability to visualize both the target and moving image in a similar latent space. This property allows us the flexibility to learn a similarity metric between their latent space encoded features, whereas the two-input channel CNN architecture earlier (Cheng et al., 2018), (Simonovsky et al., 2016) only allows a single latent space representation for the input images. Unlike DeepSim's CNN autoencoder, Siamese networks provide a more efficient way of extracting features from similar input images with their weight-sharing property (Bromley et al., 1993). Table 1 and Table 3 show that the dual-encoder architecture of SiamRegQC competitive registration performance for multimodal and unimodal datasets.

## B.2. Rationale for using a final classification task instead of regression task

Training a supervised DL network aims to learn meaningful semantic similarities and provide a numerical quality measure for the registration between a pair of images. We exploit the intermediate Siamese encodings to measure a numerical cosine similarity value as the similarity between the input target and the moving images. Since the regression task of autoencoding used to train only the encoders already learns semantic representations of a single input image in the first training stage, we use a second stage of training to provide the context of the aligned and misaligned input pairs. The classification task assists the framework to be oriented to the categorical (aligned or misaligned informed) decision-making step that increases the sensitivity of SiamRegQC to misalignment errors and minimizes the risk of losing essential misalignment information. Adding a contrastive loss (increases discrimination between aligned and misaligned pairs) that leverages these categorical labels is more straightforward in a classification framework than in a regression framework. Also, the self-supervised classification task conceptually aligns with the previous works of (Cheng et al., 2018), (Simonovsky et al., 2016). Hence, we opt for a classification task to train SiamRegQC end-to-end in the second training step.

## Appendix C. Desirable aspects of SiamRegQC for critical registration evaluation (Continued Section 3.1)

In this section, we continue our analysis of SiamRegQC as a sensitive and critical metric with some qualitative examples and registration algorithms for the section-wise registration of histological sections.

**Consistency of Evaluation Metric**- In Section 3.1, we have seen that SiamRegQC can distinguish between visibly misaligned images with increased sensitivity, even seen in Columns 1 and 2 of Figure 8. This section discusses another important aspect of an evaluation metric to maintain a consistent value for visibly well-registered images. From Columns 4 and 5 of Figure 8, we see that MSE and SSIM have largely different numerical values (inconsistent) for visibly well-registered images, while SiamRegQC can consistently evaluate them. Note that the metric values for MSE and SiamRegQC (except in the last Column) are scaled by dividing them with the value for the smallest translation misalignment error of 0.001 mm for better interpretation.

**Statistical Benchmarking Performance of 2D Rigid Registration Algorithms of Nissl-stained Histological Volume Reconstruction-** In 3D histological volume reconstruction, the reconstructed volume is formed by successively registering neighboring 2D serial sections to one another (Kajihara et al., 2017). In this instance, even small subpixel registration errors in the successive 2D section-wise registrations can accumulate over a number of sections, possibly resulting in a skewed or distorted 3D volume (Lobachev et al., 2021). While MSE and SSIM tend to overlook such small errors, SiamRegQC can pick on them with increased sensitivity, as seen in Figure 3. We consider any pair of consecutive coronal sections of the Nissl-stained mouse brain dataset as a ground truth pair of sections (Refer Appendix. D for more details). We benchmark three registration algorithms by comparing their results to the original pair of sections with the Welch two-sample t-test. A registration algorithm can be termed "good" when a mean evaluation metric of the registered pair of sections is as close to the mean metric value of the ground-

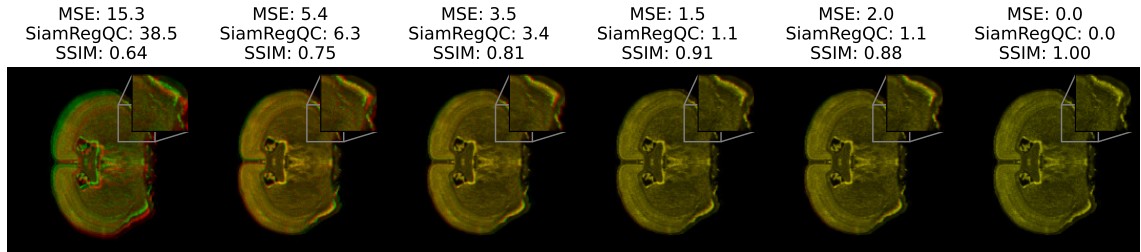

Figure 8: Evaluation of SITK registered images starting from random initial misalignment errors to demonstrate the consistency of SiamRegQC metric. Columns 1 to 3: Registered images in decreasing order of misalignment errors as seen by visual inspection. Columns 4, and 5: Visually well-registered images show inconsistent MSE and SSIM metrics, while SiamRegQC shows a consistent metric value. Last column: Ground truth registration solution for a given target image.

truth pair of sections as possible. Statistically, the p-value associated with the Welch test helps in assigning a confidence value to the "goodness" of registration. A p-value less than 0.05 suggests that the results of the registration algorithm are significantly different from the ground-truth images, indicating a low confidence value to evaluate a registered algorithm as "good." Alternatively, a p-value greater than 0.05 shows that a registration algorithm is "good" with more confidence. From Table 2, a relatively poorly performing FFT (Reddy and Chatterji, 1996) method becomes a trivial case of evaluation, which even the traditional metrics can adequately prove with higher mean differences with the ground truth, p-value $<< 0.05$ and high absolute $T_{stat}$ values of 15.4 for MSE and 10.1 for SSIM respectively. The overall well-performing SITK (Yaniv et al., 2017) and SIFT (Lowe, 2004) algorithms become non-trivial cases of evaluation. Especially for the SITK algorithm, while MSE, SSIM, and SiamRegQC$_{np\_nl}$ show that the registration outputs are "good" with a higher confidence score (p-value greater than 0.05, highlighted in red), the other variants of SiamRegQC are still critical about their confidence (highlighted in blue) of SITK being a "good" algorithm. This shows that SiamRegQC considers cases as shown in 8, Column 1 with high criticality and hence shows conservative p-value confidence about the "goodness" of the SITK algorithm. These benefits of SiamRegQC are further explored when SiamRegQC shows better registration performance than previous deep similarity metrics as seen in Section 3.2.

## Appendix D.  Evaluation of Section-wise Registration of Nissl Mouse Brain Volume Reconstruction

The process of histological volume reconstruction with section-wise 2D registrations is detailed and traditionally evaluated with metrics like MSE and SSIM, as shown by (Lobachev et al., 2021). The section-wise registration process can be summarized as follows. Consider a set of $n$ serial section histology images, $S_1$, $S_2$, ... $S_n$, to be aligned to form a 3D volume, V. We begin with the middle section of the histological stack, $S_{n//2}$ as a reference and

| Evaluation Metric | | Registration Algorithm | | | |
|---|---|---|---|---|---|
| | | Ground Truth | SIFT | SITK | FFT |
| MSE | $\mu \pm \sigma$ | $0.001 \pm 0.001$ | $0.002 \pm 0.001$ | $0.001 \pm 0.001$ | $0.007\ 0.005$ |
| | $T_{stat}$ | - | 3.7 | 0.1 | 15.4 |
| | $p_{val}$ | - | 2.6e-04 | 0.91 | 1e-36 |
| SSIM | $\mu \pm \sigma$ | $0.896 \pm 0.060$ | $0.860 \pm 0.071$ | $0.890 \pm 0.066$ | $0.782 \pm 0.099$ |
| | $T_{stat}$ | - | -3.9 | -0.65 | -10.1 |
| | $p_{val}$ | - | 1e-04 | 0.52 | 4.9e-19 |
| SiamRegQC$_{np\_nl}$ | $\mu \pm \sigma$ | $0.004 \pm 0.005$ | $0.006 \pm 0.006$ | $0.005 \pm 0.005$ | $0.114 \pm 0.105$ |
| | $T_{stat}$ | - | 4.9 | 1.2 | 15.0 |
| | $p_{val}$ | - | 1.7e-e-06 | 0.12 | 3.4e-35 |
| SiamRegQC$_{np}$ | $\mu \pm \sigma$ | $0.010 \pm 0.010$ | $0.015 \pm 0.010$ | $0.012 \pm 0.010$ | $0.163 \pm 0.127$ |
| | $T_{stat}$ | - | 5.9 | 2.3 | 17.3 |
| | $p_{val}$ | - | 7.9e-09 | 0.02 | 3.5e-42 |
| SiamRegQC$_{nl}$ | $\mu \pm \sigma$ | $0.017 \pm 0.014$ | $0.029 \pm 0.017$ | $0.023 \pm 0.013$ | $0.209 \pm 0.133$ |
| | $T_{stat}$ | - | 7.3 | 3.4 | 20.1 |
| | $p_{val}$ | - | 2.7e-12 | 2.2e-04 | 3.2e-53 |
| SiamRegQC | $\mu \pm \sigma$ | $0.024 \pm 0.017$ | $0.039 \pm 0.023$ | $0.031 \pm 0.021$ | $0.264 \pm 0.145$ |
| | $T_{stat}$ | - | 7.2 | 4.0 | 23.7 |
| | $p_{val}$ | - | 1.9e-12 | 7.6e-05 | 7.9e-62 |

Table 2: Welch two-sample T-test between registration algorithms and ground truth images. MSE and SSIM rate SITK as a 'good' registration method with a p-value greater than 0.05, while SiamRegQC provides a statistically significant evaluation with a p-value greater than 0.05.

perform pairwise registrations of serial sections as given below:

$$\text{Aligned } S_i = \begin{cases} T(S_i, S_{i-1}), & \text{if } i \geq n \\ T(S_i, S_{i+1}), & \text{if } i < n \end{cases} \tag{5}$$

where, T is the rigid transform that registers $S_i$ to $S_{i\pm1}$.

In Section C and Table 2, pairs ($S_i$, $S_{i\pm1}$) from the Nissl dataset are considered as the ground-truth sections. We synthetically misalign successive serial sections as described in Figure 9 and use three different registration algorithms viz., featured-based (SIFT) (Lowe, 2004), intensity-based (SITK) (Yaniv et al., 2017), and FFT-based registration (Reddy and Chatterji, 1996) to register them together. The experimental results and performances of the mentioned algorithms are detailed in Section C.

## Appendix E. SiamRegQC as a deep similarity metric for unsupervised non-rigid deformable registration for unimodal images

In this section, we evaluate the effectiveness of SiamRegQC on a more complex downstream task of non-rigid deformable registration. We first describe the process of generating synthetic non-rigid misaligned images for training SiamRegQC. Later, we study the effect of using SiamRegQC as a similarity metric to drive unsupervised non-rigid registration in comparison with previous deep similarity metrics for unimodal data.

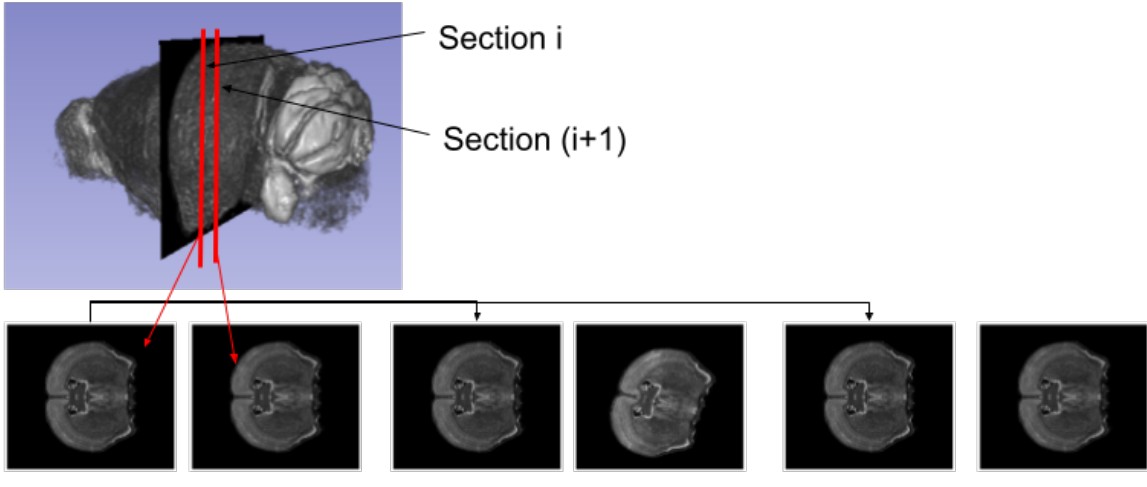

Figure 9: Ground truth pairwise sections extracted from the original 3D Nissl-stained volume. The sections are synthetically misaligned and registered back using three different algorithms: SIFT, SITK, and FFT. SiamRegQC is used to benchmark the algorithms critically.

### E.1. Generation of Non-Rigid Deformable Transformations

To simulate non-rigid misaligned images for training SiamRegQC, we generate random flow-field grids and resample the input image with bilinear interpolation to get non-rigid deformed images, as shown in Figure 10. The generated random flow fields are smoothened and scaled randomly with parameters $\alpha$ and $\sigma$, respectively. We train SiamRegQC with original and synthetically created misaligned pairs of images as described in Section 2. We use about 3000 T1 MRI sections from the same IXI dataset described in Section 2.4 for training SiamRegQC with non-rigid misaligned and aligned images.

### E.2. Comparison with previous Deep Similarity Metrics

To test the effectiveness of SiamRegQC beyond rigid registration and quality evaluation as described in Section 3, we study the effects of leveraging SiamRegQC as a similarity metric (interchangeably referred to as loss function) to drive unsupervised non-rigid deformable registration. We use a VoxelMorph (Balakrishnan et al., 2019) architecture for setting up a learning-based unsupervised deformable registration framework, as shown in Figure 5. We compare SiamRegQC with other deep similarity metrics like DeepSim (Czolbe et al., 2023) and the two-channel CNN-based metric proposed by (Cheng et al., 2018). More details on these metrics and their methodological differences with SiamRegQC are discussed in Appendix A, B.

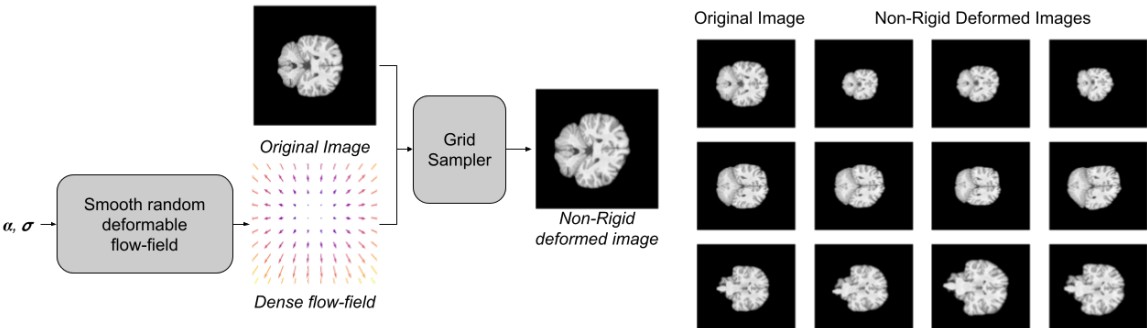

Figure 10: Left: Graphical representation of synthetically generating non-rigid misaligned images. $\alpha$ and $\sigma$ denote the parameters used to control the scale and smoothness of the deformed (misaligned) grid. Right: Examples of generated non-rigid misaligned images.

| Evaluation Metric | Deformed Image (Before Registration) | VXM$_{\text{mse}}$ | VXM$_{\text{ncc}}$ | VXM$_{\text{Cheng}}$ | VXM$_{\text{DeepSim}}$ | VXM$_{\text{SiamRegQC}}$ | ANTsPy |
|---|---|---|---|---|---|---|---|
| MSE | $0.016 \pm 0.017$ | $0.0027 \pm 0.007$ | $0.0013 \pm 0.006$ | $0.0024 \pm 0.007$ | $0.0023 \pm 0.006$ | $0.0011 \pm 0.005$ | $0.0010 \pm 0.004$ |
| SSIM | $0.699 \pm 0.134$ | $0.935 \pm 0.127$ | $0.962 \pm 0.123$ | $0.920 \pm 0.124$ | $0.936 \pm 0.127$ | $0.967 \pm 0.109$ | $0.987 \pm 0.031$ |

Table 3: Quantitative Evaluation of SiamRegQC before and after registration with other traditional and deep similarity metrics. Considering the non-learning-based ANTsPy as a reference registration method, SiamRegQC shows the closest performance to ANTsPy (highlighted in blue).

Table 3 shows the quantitative evaluation that SiamRegQC performs better than other traditional and deep similarity metrics with a mean SSIM value of 0.967 (highlighted in blue), closely followed by the Normalized Cross Correlation (NCC) optimized Voxel-Morph network. From Figure 11, ANTsPy shows the least amount of dispersion and data points close to 0 for MSE and close to 1 for SSIM. From the learning-based VoxelMorph networks trained with different loss terms, SiamRegQC shows the closest dispersion to ANTsPy. Although the non-learning-based traditional ANTsPy (Avants et al., 2009) registration method shows the best MSE and SSIM metrics, ANTsPy works with an average inference time of 3 minutes for every registration run. Whereas VXM$_{\text{SiamRegQC}}$ records an average inference time of 0.18 seconds for a single registration run. The qualitative results supporting SiamRegQC's superior performance are described in Figure 12.

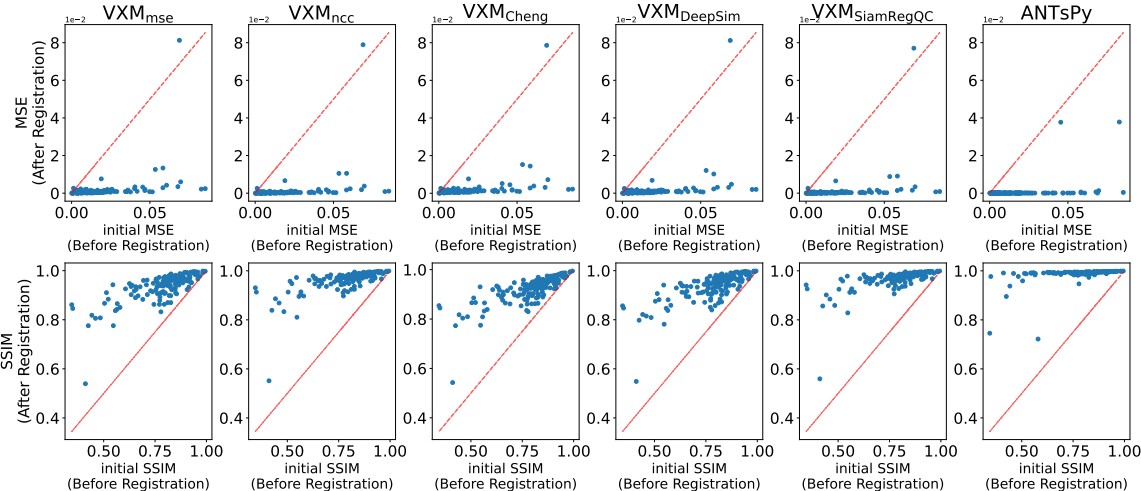

Figure 11: (Top row) Improvement in MSE metrics after using SiamRegQC as a similarity-based loss function for unsupervised deformable registration. Each data point represents the evaluation metric for one pair of registered and target images. The dashed line represents identity transformation. All data points below the dashed line suggest improvement in the registration performance. (Bottom row) Improvement in SSIM metrics after using SiamRegQC as a similarity-based loss function for unsupervised deformable registration. All data points above the dashed line suggest improvement in the registration performance. Each registration method name is denoted as VXM, with the subscript indicating the type of similarity metric used as the loss function. For instance, $\mathrm{VXM_{mse}}$ denotes VoxelMorph method with MSE as the loss function.

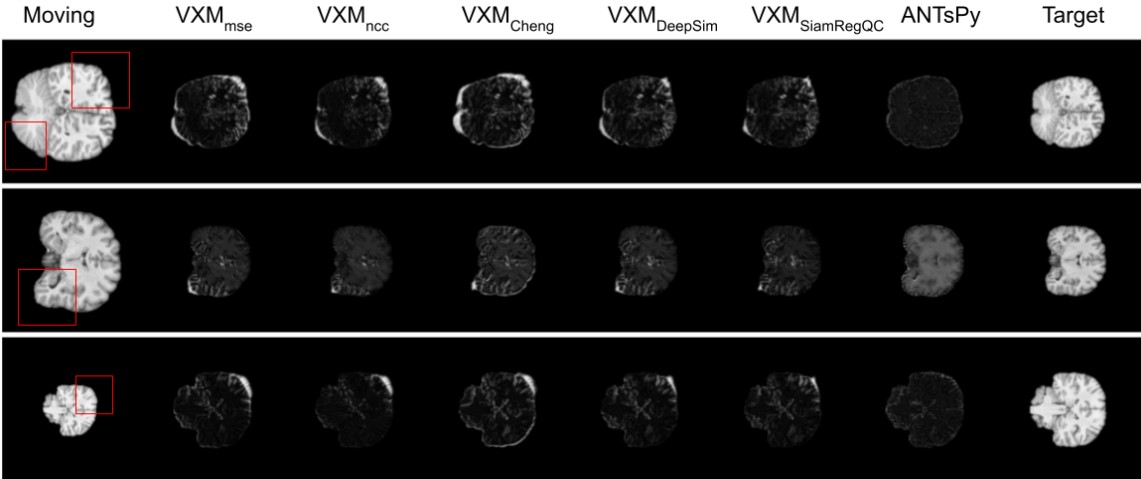

Figure 12: Registered images from different VoxelMorph networks supervised with various traditional and deep-similarity-based loss functions. Except for the input target and moving images, the VoxelMorph outputs are displayed as differences from the target images for better visualization of the registration error.

