# OpenReview forum: "Registration Quality Evaluation Metric with Self-Supervised Siamese Networks"
_MIDL.io/2024/Conference — MIDL 2024 Poster_

### Official Review · Reviewer_DCSe · 2024-02-26

**Confidence:** 4
**Preliminary Rating:** 2
**Recommendation:** Poster
**Final Rating:** 3.5

**Summary:**

This paper illustrates the benefits of using data-driven similarity metrics for evaluating rigid registration.
Similarity metrics were calculated as the cosine similarity of CNN features from two inputs (target & registered). CNNs were trained with cross-entropy from synthetic misalignment labels (aligned or not) and contrastive loss.
Three main experiments were shown - smoothness of evaluation, visual consistency, ability to benchmark different algorithms.
Two 2D datasets were used: 200 mouse brain histology sections and 175 human MRI coronal sections from 5 patients.

**Strengths:**

The main motivation of trying to find better evaluation metrics for image registration is clear.
The comparisons of the propose method(s) against MSE and SSIM are thorough.
The ideas behind the conducted experiments are interesting, although many of them only seemed feasible b/c the problem was restricted to rigid registration.

**Weaknesses:**

Some unclear descriptions, leading to uncertainties in understanding/evaluating the paper (details in the comments below).
Main comparisons seem to be limited to MSE and SSIM. Comparisons against more recent data-driven similarity metrics are either missing or not clearly communicated.
Method presentation and evaluation is limited to rigid registration. The proposed loss also seems to be very binary (classification and contrastive losses based on aligned vs. not aligned), which I'm not sure would translate well to nonrigid registration.

**Detailed Comments:**

In general, the paper needs several clarifications.

The main contribution for this paper seems to be not presented clearly. This was not the first paper to propose CNN-based evaluation metrics for registration, since Czolbe et. al (2023) proposed a very similar approach using cosine similarity of CNN encoder features & training of CNN with relevant tasks.
The main difference in this paper seems to be that it uses a different training procedure to obtain better CNN features for evaluation. This point needs to be more clearly communicated.

Similarly, the comparisons between the current approach and previous CNN-based evaluation metrics need to be clarified. From my understanding, SiamRegQC_{np_nl} is similar to Czolbe et. al (2023) with just the autoencoder training, since np_nl seems to suggest no training with equation (1). However, this point is not very clear because no direct connections are made, and it is difficult to understand what is meant by the "no-prior" condition (what is the data-specific prior? there's no description).

If the above is not a correct analogy, then the bigger issue is that there are no comparisons against previous CNN-based evaluation metrics.

In Table 1, what is the meaning of the ground truth column? It's under the "Registration Algorithm" header, so does that mean it's actually supposed to be another evaluation of a registration technique? Need clearer explanations for this.

Is the main takeaway from Table 1 & its Results descriptions that SiamRegQC_{np_nl} is more critical of what qualifies as a "good" algorithm? How can we evaluate if this is an accurate assessment? (i.e. why is it better to label SITK as a "bad" algorithm?)

"Triplet contrastive loss" - it would be good to at least provide a citation for the exact mathematical definition.

**Justification Of Final Rating:**

The manuscript has been significantly altered based on the reviewer comments.
The experimental results seem to be much more aligned with the reviewers' expectations.
Methods have been clarified based on the reviewer comments.
The quality of the initial submission, however, needs to be taken into account -- hence the final rating of borderline accept.

**Justification Of The Preliminary Rating:**

Evaluation techniques are interesting, but..
There are multiple areas that need better explanations and/or citations.
Comparisons to previous deep learning-based image similarity metrics may be missing (or not clearly communicated).
The method is limited to 2D rigid registration.

**Questions To Address In The Rebuttal:**

All of my major concerns are listed in the detailed comments. I believe carefully addressing those would be great for the rebuttal.

**Special Issue:**

No

---

> ### Author Response · Authors · 2024-03-15
> **Addressing specific reviews**
>
> Thank you for highlighting some crucial aspects required to improve the quality of our work. Here are some of the concerns we have identified from the reviews to the best of our understanding, and we hope to clarify them with all sincerity.
>
> __Method evaluation is limited to rigid registration__
>
> We test the applicability of SiamRegQC as a deep similarity metric to drive the VoxelMorph registration framework for unimodal and T1-T2 multimodal IXI data (Appendix A). SiamRegQC offers the best registration performance with an average SSIM score of 0.967 and 0.845 for unimodal and multimodal data, respectively.
>
> __Main comparison limited to MSE and SSIM. Comparisons against more recent similarity metrics are either missing or not clearly communicated__
>
> We compare SiamRegQC with previously proposed deep similarity metrics, DeepSim, and another CNN-based metric proposed by Cheng et al. in the VoxelMorph registration experiment (Please refer to Appendix A2).
>
> __Unclear explanations of the methodology of SiamRegQC and its main contribution__
>
> We use Siamese Networks to encode the input target and moving images into latent space with a two-step training process.
> (i)  The encoder, $\phi$ of SiamRegQC, is trained with an unsupervised image reconstruction task on a single MRI image.
> (ii)  The entire network (encoder + FCN) is trained end-to-end on a classification task based on the categorical labels (aligned or misaligned) assigned to a pair of target and moving images (Kindly refer to Figure 11 and Appendix B for more details of SiamRegQC architecture).
>
> We appreciate your understanding of drawing parallels between DeepSim and the baseline variants of SiamRegQC. While SiamRegQC is arguably similar to DeepSim in the first training step, a significant difference between DeepSim and SiamRegQC is that DeepSim is only trained on a single input image reconstruction task. SiamRegQC is trained with the pair of target and moving images in a misalignment classification task context. This may be why SiamRegQC performance is superior as a deep similarity metric for VoxelMorph registration. Using a classification task is conceptually aligned with the previous works of Cheng et al. (Refer to Appendix C for a detailed rationale of using Siamese networks for efficient feature extraction and using a classification task).
>
> The rationale for comparing SiamRegQC with $\mathrm{SiamRegQC_{np}}$, $\mathrm{SiamRegQC_{nl}}$ and $\mathrm{SiamRegQC_{np\_nl}}$ is to perform an ablative study to highlight the importance of adding a contrastive loss and pre-training of the encoder, $\phi$, for more sensitivity to smaller errors. Here, ‘np’ indicates no pre-training (revised from ‘no-prior’), and ‘nl’ means lack of additional contrastive loss in the classification training step (only Cross Entropy Loss is used). Section 2.2 is modified accordingly.
>
> __Proposed loss seems to be very binary (only classification and contrastive loss)__
>
> The encoder, $\phi$, is already pre-trained to learn spatial semantic features of the input data. The classification task assists the framework in being oriented to the categorical (aligned or misaligned informed) decision-making step that increases the sensitivity of SiamRegQC to misalignment errors and minimizes the risk of losing essential misalignment information.
>
> __In Table 1, what is the meaning of ‘ground truth’ column?__
>
> In Table 1, ground truth sections refer to the successive serial sections of the original 3D Nissl-stained volume of the mouse brain. Appendix D details the process of section-wise registrations and the creation of synthetic misalignments. The synthetically misaligned sections are registered with three registration algorithms and benchmarked with the original sections (the ground truth).
>
> __Triplet Contrastive Loss is not well-defined and cited__
>
> We have expanded the loss function used in Equation 1 to elucidate the formulation of contrastive loss (modified the one proposed by Chen et al.). The contrastive loss uses categorical labels (aligned or misaligned)  to discriminate between the latent encodings of our input categories - aligned and misaligned. Thereby helping SiamRegQC to attain better sensitivity to more minor misalignment errors as observed in Figure 2 and Section 2.2.
>
> For better space management, we have continued our rebuttal for the main takeaway from Table 1.
>
> We kindly request you to check our common rebuttal responses and manuscript changes, which mention the details of the additional experiments performed for multimodal non-rigid registration using the VoxelMorph registration framework. Once again, we are deeply grateful for your genuine reviews and hope to have addressed all the concerns raised to your satisfaction. Kindly refer to all our updates and reconsider improving the ratings based on our updates.

---

> ### Author Response · Authors · 2024-03-15
> **Main takeaway from Table 1**
>
> __What is the main takeaway from Table 1?__
>
> Unfortunately, we did not present the nuances of the p-value of the Welch T-test in evaluating a registration algorithm as ‘good’ or ‘bad.’ However, we want to take this opportunity to rephrase our interpretations of Table 1 and clarify our notions of a ‘bad’ algorithm.
> The mean metric value of a registration algorithm outputs is as close to the original sections (ground truth) as possible, suggesting that the registration algorithm is ‘good.’ The p-value provides a confidence measure to show if there is significant evidence that the registered outputs differ from the ground truth sections. If the p-value $>$ 0.05, there is insufficient evidence that the registered outputs differ from the ground truth (high confidence that the registration algorithm is ‘good’). Alternatively, a p-value $<<$ 0.05 shows that despite the mean metric values being close to the ground-truth sections, there is sufficient evidence that the registered outputs are statistically different from the ground-truth sections (low confidence that the registration algorithm is ‘good’). Table 1 suggests that MSE, SSIM, and $\mathrm{SiamRegQC_{np\_nl}}$ show that the challenging case of the SITK registration algorithm is ‘good’ with high confidence. However, SiamRegQC is critical of the ‘goodness’ of the algorithm with a p-value $<$ 0.05 (low confidence). With examples shown in Figure 4, SiamRegQC can better penalize smaller misalignments, even demonstrated by the sensitivity plots in Figure 2.  Please refer to Section 3.4 for a similar rephrased interpretation of the results from Table 1.
>
> We kindly request you to check our common rebuttal responses and manuscript changes, which mention the details of the additional experiments performed for multimodal non-rigid registration using the VoxelMorph registration framework. Once again, we are deeply grateful for your genuine reviews and hope to have addressed all the concerns to your satisfaction. Kindly refer to all our updates and reconsider improving the ratings based on our updates.

---

> > ### Comment · Reviewer_DCSe · 2024-03-18
> >
> > Thank you to the authors for carefully addressing the comments.
> > 1. The paper is a bit clearer now, but I still think something like Figure 11 belongs in the main manuscript. I still cannot find info on pre-training based on "an unsupservised image reconstruction task" in the main manuscript. Also, what is unsupervised image reconstruction? Autoencoding? What loss is used for pre-training? This is a vital component of the algorithm, and arguably the main differentiating contribution -- it should not be omitted.
> > 2. From my understanding, Section 3.4 and Table 1 is basically saying, if the p-value is low, the metric is still critical of the registered images and **will further improve the registration if we use that metric for optimization.**  As of now, the significance of this section is unclear.
> > 3. Even with the added statement, I'm not sure Section 3.4 is more beneficial for the paper than some of the content in the appendix (e.g. nonrigid registration with comparisons with deep learning evaluation metrics). In general, the additions are good, but the main paper would need to be significantly restructured to be a good standalone paper.
> > 4. Based on the equations, I believe the correct wording should be "contrastive loss", not "triplet contrastive loss." Triplet requires an anchor, positive, and negative sample.
> > 5. Quotations should use `{text}'. Currently, all quotation marks on the left hand side are facing the wrong direction.

---

> ### Author Response · Authors · 2024-03-18
> **Restructuring manuscript to highlight multimodal registration with SiamRegQC as a similarity metric**
>
> Thank you for your response and reading through our updates. We appreciate your suggestions about restructuring our paper and contemplated doing it. We have revised the order of sections, keeping in mind that the multimodal registration performance with SiamRegQC is a more significant contribution than the statistical significance aspect of registration evaluation.
>
> 1. Figure 11 is added to the main paper (now Figure 2) with descriptions of the training process. By unsupervised image reconstruction, we mean the task of autoencoding, where the MSE loss between the input and decoded images is minimized. (Appropriate changes done in Section 2.2)
>
> 2.  We agree with your understanding that with a low p-value, the metric is still critical of the registration algorithm, further improving the registration optimization when used as a similarity metric. We have swapped this section with multimodal registration driven by SiamRegQC to highlight the latter contribution better.
>
> 3. We have changed the order of the Appendix in line with the general flow of the paper. Please find the updated order of the Appendix below.
>
>     A- Previous related works
>
>     B- Rationale for the architecture of SiamRegQC
>
>     C- Continued desirable aspects of SiamRegQC
>
>     D- Section-wise Registration evaluation for Nissl-stained volume reconstruction
>
>     E- SiamRegQC as a deep similarity metric for VoxelMorph registration of unimodal images.
>
> 4. Apologies for the use of the word 'triplet'. We do not have an anchor, and hence it is corrected.
>
> 5. We have rectified the quotation marks.

---

### Official Review · Reviewer_Xd7o · 2024-02-28

**Confidence:** 4
**Preliminary Rating:** 2
**Recommendation:** Poster
**Final Rating:** 4

**Summary:**

The paper introduces SiamRegQC, a novel metric for measuring registration quality, akin to classical metrics like MSE or SSIM. SiamRegQC assigns a scalar value to a pair of images. To calculate this score, each image is processed through the same encoder network to generate a feature vector. The SiamRegQC score is derived by computing 1 minus the similarity score, which is based on the normalized correlation between the two feature vectors. The encoder network is trained using a self-supervised scheme, where synthetic pairs of images, created through rigid transformations, are classified as "aligned" or "misaligned" by an auxiliary network. The paper conducts qualitative and quantitative experiments to explore the behavior of SiamRegQC in terms of smoothness and minima and compares it with MSE and SSIM.

**Strengths:**

The introduction of a reliable metric for measuring similarity between image pairs is particularly crucial for unsupervised image registration. Traditional metrics like MSE, SSIM, or cross-correlation often fall short in multimodal scenarios. Network-based metrics, such as the one proposed, could potentially overcome these limitations. Therefore, this approach is highly relevant and could have a significant impact in the field.

**Weaknesses:**

The paper lacks crucial information, leaving ambiguities about its methodologies. For instance, the use of contrastive loss is mentioned but not elucidated, leaving readers to infer its application and significance. Equation 2, presumably representing the normalized dot product between feature vectors, is unclear due to ambiguous notation. Clarification here would be beneficial. The choice of a classification approach over a regression model during training is unexplained. Additionally, the evaluation of the metric seems limited by the small dataset used. A more exhaustive exploration, possibly by integrating SiamRegQC into unsupervised registration methods like Voxelmorph and/or demonstrating its utility in multimodal image registration, would strengthen its validation.

**Detailed Comments:**

Refer to "Weaknesses" for suggestions on minor improvements or clarifications.

**Justification Of Final Rating:**

Thank you very much for revising the manuscript and addressing my concerns. I am pleased to see that the training of Voxelmorph was successful. Consequently, I will raise my rating to 'weak accept.' Thank you!

**Justification Of The Preliminary Rating:**

The paper presents insufficient detail on crucial aspects, particularly regarding methodologies and their underlying rationales. Furthermore, the evaluation of the proposed metric appears limited, lacking comprehensive experiments that demonstrate its effectiveness in practical scenarios, such as unsupervised image registration. The inclusion of such experiments is essential to ascertain the utility and applicability of the proposed methods in advancing the field.

**Questions To Address In The Rebuttal:**

(1) Is Equation 2 calculating the normalized dot product between the two feature vectors?
(2) What was the rationale behind opting for a classification approach rather than a regression problem during training?
(3) Has there been any attempt to utilize the metric for optimizing an image registration problem?

**Special Issue:**

No

---

> ### Author Response · Authors · 2024-03-15
> **Addressing specific concerns  (Updated restructured references)**
>
> Thank you for highlighting crucial aspects required to improve the quality of our work. Here are some of the concerns we have identified from the reviews to the best of our understanding, and we hope to clarify them in all sincerity.
>
> __Missing details in the methodology sections__
>
> We use Siamese Networks to encode the input target and moving images into latent space with the help of a two-step training process.
> (i)  The encoder, $\phi$, is trained with an unsupervised image reconstruction task on a single MRI image.
> (ii)  The entire network (pretrained encoder + FCN) is trained end-to-end on a classification task based on the categorical labels (aligned or misaligned) assigned to a pair of target and moving images.  (_Figure 2 and Appendix A, B_ )
>
> __Evaluating the performance on a large dataset__
>
> We train SiamRegQC on 4000 coronal sections of the IXI MRI dataset and evaluate another  200 sections for the VoxelMorph experiment.
>
> __Integration of SiamRegQC into unsupervised registration methods like VoxelMorph__
>
> We test the efficacy of SiamRegQC by training a VoxelMorph registration using SiamRegQC as a loss function.
> SiamRegQC shows the best registration performance (closest to traditional ANTsPy registration) evaluated with SSIM (0.967) and MSE (0.0011), compared with other similarity metrics. Kindly refer to _Section 3.2, _Appendix A_ for more details.
>
> __Utility in multimodal registration__
>
> We perform the VoxelMorph registration on the T1-T2 MRI images to demonstrate the utility of SiamRegQC on a multimodal dataset. SiamRegQC performs competitively with other multimodal similarity metrics and achieves an average SSIM score of 0.845 and NCC score of 0.825. Kindly refer to  _Section 3.2_ for more details.
>
> __Rationale behind using classification over regression during training__
>
> Since the regression task of unsupervised image registration already learns semantic representations of a single input image in the first training stage, we use a second stage of training to provide the context of the aligned and misaligned input pairs.
> The classification task assists the framework to be oriented to the categorical (aligned or misaligned informed) decision-making step that increases the sensitivity of SiamRegQC to misalignment errors and minimizes the risk of losing essential misalignment information.
> Adding a contrastive loss (increases discrimination between aligned and misaligned pairs) that leverages these categorical labels is more straightforward in a classification framework than in a regression framework.
> Hence, we opt for a classification task to train SiamRegQC end-to-end. On another note, using a regression task in this second stage of training would require the formulation of a traditional metric to regress to and complicate the training procedure. Appendix C in the manuscript details this rationale for the benefit of all readers.
>
> __Use of contrastive loss is not elucidated__
>
> We have expanded the loss function used in Equation 1 to elucidate the formulation of contrastive loss. The contrastive loss uses categorical labels to discriminate between the latent encodings of our input categories - aligned and misaligned, thereby helping SiamRegQC attain better sensitivity to more minor misalignment errors, as observed in _Figure 3_ and Section 2.2.
>
> To specifically answer the three questions raised in the review, we reiterate some of the above explanations.
>
> __Questions to address in rebuttal:__
>
> _Is Equation 2 calculating the normalized dot product between the two feature vectors?_
>
> -Yes. The normalized dot product is called cosine similarity function in the paper and formulated in Equation 3.
>
> _What was the rationale behind opting for a classification approach rather than a regression problem during training?_
>
> -Our method learns from a combination of a two-step training process that uses both regression and classification. The classification task is chosen in the second step for a straightforward implementation of the contrastive loss essential for increasing the sensitivity of SiamRegQC to smaller errors. More details are explained above.
>
> _Has there been an attempt to utilize the metric for optimizing an image registration problem?_
>
> -We have incorporated SiamRegQC as a loss function in unsupervised VoxelMorph registration framework and added this experiment in _Section 3.2_. We observed that SiamRegQC achieves the best average SSIM score of 0.967 over previously proposed deep similarity-based and traditional metrics.
>
> We kindly request you to check our common rebuttal responses and manuscript changes, which mention the details of the additional experiments performed for multimodal non-rigid registration using the VoxelMorph registration framework. Once again, we are deeply grateful for your genuine reviews and hope to have addressed all the concerns raised to your satisfaction. Kindly refer to all our updates and reconsider improving the ratings based on our updates.

---

> > ### Comment · Reviewer_Xd7o · 2024-03-27
> >
> > Thank you very much for revising the manuscript and addressing my questions. I have no further questions at this time. I am pleased to see that the training of Voxelmorph was successful.

---

> > > ### Author Response · Authors · 2024-03-27
> > >
> > > Thank you for your feedback and for acknowledging our revisions.

---

### Official Review · Reviewer_o46a · 2024-02-29

**Confidence:** 4
**Preliminary Rating:** 1

**Summary:**

This paper proposes a deep similarity metric to evaluate the alignment between fixed and moving image pairs for multimodal image registration and is applied to histo and MRI rigid slice registration. The idea is to train an autoencoder $\phi$ to learn a feature extractor (encoding path of the encoder-decoder) that is then applied to aligned or unaligned images, the features of which are then used to classify between aligned and unaligned images using a two class classifier. This process is done end-to-end.
Once trained, the cosine similarity between $\phi(fixed)$ and $\phi(moving)$ is used as a similarity metric. The method is validated on 2D histological brain images and 2D MRI images and compared to MSE and SSIM using rigid registration only. Results show better local minima for the proposed metric over MSE or SSIM.

**Strengths:**

The paper proposes a new deep similarity metric that would overcome the inherent limitations of current metrics, which is an important topic.
The paper is reasonably fluent, which makes its contribution clear enough, although the method description is short.

**Weaknesses:**

The only comparison is to MSE or SSIM in the context of rigid registation. This comparison is largely insufficient as many other metrics would show better convergence profiless  (e.g. MIND, NGF to name the two most famous contenders.) This  disqualifies the paper in my opinion, and it shows the lack of positioning with respect to the image registration literature and progresses made over the last 20 years.

Perceptual deep registration metrics have been proposed in the literature for more than 8 years now (e.g. seminal works of Simonovski Mateus et al. MICCAI 2016). These are not or loosely discussed in this paper, and not evaluated. The application (rigid registration of 2D images using simulated translations and rotations) is obsolete when compared to current advances in image registration.

**Detailed Comments:**

claims of superiority over MSE or SSIM are obsolete in the context of image registration, which disqualifies the paper in my opinion.

**Justification Of The Preliminary Rating:**

This paper, while adressing the important topic of identifying better registration metrics for deep unsupervised image registration, fails at convincing. Its positioning with respect to state of the art is fundamentally flawed, by using MSE and SSIM as its sole competitors in the context of image registration. I therefore recommend rejection.

**Questions To Address In The Rebuttal:**

comparisons with relevant deep and conventional similarity metrics + other applications...

**Special Issue:**

No

---

> ### Author Response · Authors · 2024-03-16
> **Addressing specific concerns (Updated restructured references)**
>
> Thank you for highlighting some crucial aspects required to improve the quality of our work. Here are some of the concerns we have identified from the reviews to the best of our understanding, and we hope to clarify them with all sincerity.
>
> __Evaluation restricted to rigid registration__
>
> We perform a new experiment where we extend the applicability of SiamRegQC as a deep similarity metric to drive the VoxelMorph registration framework for TI MRI data and T1-T2 multimodal IXI data (https://brain-development.org/ixi-dataset/). The experimental setup and results are detailed in _Appendix E_. We observe that SiamRegQC offers the best registration performance with an average SSIM score of 0.967 and 0.845 for unimodal (_Appendix E_) and multimodal data (_Section 3.2_), respectively.
>
> __Comparison with previous deep similarity metrics is missing__
>
> We compare the effectiveness of SiamRegQC with previously proposed deep similarity metrics DeepSim and another CNN-based metric proposed by Cheng et al. (Main cited paper in Simonovosky 2016 and similar architecture) in the VoxelMorph registration experiment. More details are presented in _Section 3.2, Appendix E_.
>
> __Literature survey is not detailed, and the method description is short__
>
> We mainly compare our proposed metric, SiamRegQC, with previously proposed DeepSim and a CNN-based metric by Cheng et al. We combine the training methodologies from these previous metrics and propose an enhanced SiamRegQC metric using Siamese networks to compute the cosine similarity score (Please refer to  _Appendix A and B_ for a detailed explanation).
> We use Siamese Networks to encode the input target and moving images into latent space with the help of a two-step training process.
> (i)  The encoder, $\phi$ of SiamRegQC, is trained with an unsupervised image reconstruction task on a single MRI image.
> (ii)  The entire network (encoder + FCN) is trained end-to-end on a classification task based on the categorical labels (aligned or misaligned) assigned to a pair of target and moving images. Kindly refer to _Figure 2 and Appendix A, B_ for details on the architecture for training SiamRegQC.
>
> __Other applications are not explored__
>
>  We extend the applicability of SiamRegQC as a deep similarity metric to drive the VoxelMorph registration framework for TI MRI data and T1-T2 multimodal IXI data (_Section 3.2_). The Siamese architecture with the two-step training method for SiamRegQC performs competitively with the other deep similarity metrics.
>
> We are currently working on and trying our best within the available rebuttal time to compare SiamRegQC with traditional metrics like MIND and ngf for multimodal datasets as suggested in the reviews. We kindly request you check our common rebuttal responses and manuscript changes, which mention the details of the additional experiments performed for multimodal non-rigid registration using the VoxelMorph registration framework. Once again, we are deeply grateful for your genuine reviews and hope to have addressed most concerns to your satisfaction. Kindly refer to all our updates and reconsider improving the ratings based on our updates.

---

> ### Author Response · Authors · 2024-03-17
> **Comparison with conventional multimodal metrics  (Updated restructured references)**
>
> We have added the comparison of SiamRegQC with the conventional multimodal similarity metric called MIND (Heinrich et al. 2022). Please refer to _Section 3.2_ for more details. We observe that SiamRegQC achieves a superior NCC and SSIM score of 0.825 and 0.845 over MIND, respectively.

---

### Author Response · Authors · 2024-03-14
**Thanking the reviewers and addressing common concerns (Updated restructured references)**

Thank you for your helpful reviews. We appreciate your feedback and are motivated to address all the genuine concerns raised to improve the quality and impact of our research. We would like to take this opportunity to clarify all ambiguities and give our best efforts to improve the ratings so that the overall quality of the work is further enhanced. We have identified the most commonly raised concerns by all the reviewers and attempted to clarify them with a standard set of experiments. These major experiments are added in the Appendix with appropriate section headers. The main manuscript is also revised to maintain coherence with the additional experiments that are added in the Appendix. All the accompanying text revisions in the manuscript are highlighted in violet color. The following are identified concerns and our corresponding rebuttal experiments/responses.

__SiamRegQC only demonstrated on Rigid registration.__

Appendix A addresses the application of SiamRegQC for non-rigid deformable registrations.
(_Section 3.2 for multimodal data, Appendix E for unimodal data and Figure 10_) outline synthetically generating non-rigid misaligned images.

__SiamRegQC not applied to more challenging unsupervised registration tasks like VoxelMorph__

_Section 3.2_ uses SiamRegQC as a deep similarity metric-based loss function to drive unsupervised VoxelMorph registration.
_Figure 5_ depicts the utility of SiamRegQC as a loss function in a VoxeMorph registration framework.
The SiamRegQC-driven VoxelMorph network shows promising registration performance in qualitative and quantitative evaluation over previously proposed deep similarity metrics, illustrated in _Figures 6, 7, and Table. 1_.

__Comparison with previous CNN-based metrics is not mentioned (or not clearly explained)__

_Section 3.2_ compares the performance of the unsupervised VoxelMorph registration framework with two previous CNN-based similarity metrics: (i) Proposed by Cheng et al. and (ii) DeepSim, proposed by Czolbe et al.
Appendix B outlines the above metrics in more detail.

__Utility in multimodal Image Registration__

_Section 3.2_ discusses the multimodal deformable registration of MRI T1 to T2 images from the IXI dataset. Figures 6, 7, and Table. 1 show the qualitative and quantitative evaluation with commonly used traditional multimodal loss function; Normalized Cross Correlation (NCC) and MIND; two earlier DL metrics (Cheng et al. and DeepSim); and our proposed metric, SiamRegQC.
We observe that SiamRegQC can retain the sharpness and texture of T1 images after the multimodal registration.

__Previous CNN-based metrics are loosely discussed__

Appendix B discusses mainly two previously proposed CNN-based deep learning metrics that we have evaluated for comparison with our proposed metric, SiamRegQC.
_Figure. 2_ depicts the architectural differences of previous networks from SiamRegQC.

__Method Description is short, and the rationale for choosing a classification task over a regression task is not discussed__

_Appendix B_ discusses the rationale for our chosen method.
Training a supervised DL network aims to learn meaningful semantic similarities and provide a numerical quality measure for the registration between a pair of images. We exploit the intermediate Siamese encodings to measure a numerical cosine similarity value as the similarity between the input target and the moving images. As the Siamese encoders are pre-trained with an unsupervised image reconstruction task, adding a regression task as the final stage is redundant. While unsupervised pre-training might not have the context of misalignment, we introduce a triplet contrastive loss that inputs binary labels (aligned vs. misaligned) and supervises a classification task. On another note, classification simplifies the methodology required to formulate an appropriate similarity measure to regress to, conceptually aligning with the previous works by Cheng et al. and Simonovosky et al.

__Contrastive Loss Function is not explained in detail__

Expanded the loss functions in the Section. 2.2. (highlighted in yellow) and provided a citation for the triplet contrastive loss used.
_Figure 3_ also serves as an ablative study to show that adding the triplet contrastive loss is essential for SiamRegQC to be most sensitive to smaller translation and rotation errors.

__Manuscript changes__

After incorporating SiamRegQC as a similarity-based loss function to drive a DL-based non-rigid registration framework for multimodal data, we have updated the abstract, contributions, and conclusion accordingly.

We would like to subsequently release our working code repository to the readers for reproducibility.  While these are only the common concerns addressed, we will provide detailed individual responses to each reviewer. We have attempted to address almost all review comments. We request you to kindly check these updates and reconsider the ratings, if the experiments are convincing enough.

---

### Author Response · Authors · 2024-03-17
**Publishing working github repository**

Respected Reviewers,

Please find the updated code repository at https://github.com/TanviKulkarni07/SiamRegQC/tree/main for training and testing SiamRegQC- registration quality evaluation metric with Siamese networks.

Please find the revised paper with experiments for unsupervised multimodal registration using SiamRegQC added in Section 3.2.

---

### Author Response · Authors · 2024-03-18
**Summary of the changes in the rebuttal version (Updated restructured references)**

Dear Reviewers,

The changes in the rebuttal version of the manuscript in line with all the reviews are summarized as follows.

__1. Application to non-rigid registration__

_Section 3.2 (Appendix  E for unimodal images)_ uses SiamRegQC to evaluate the registration of non-rigid, smoothly deformed images.

__2. Effectiveness on other applications__

_Section 3.2 (Appendix  E for unimodal images)_ details the application of SiamRegQC as a similarity-based loss function to drive Voxelmorph deformable registration.

__3. Comparison with previous deep CNN-based metrics__

_Section 3.2_ shows the results of SiamRegQC compared to previous deep similarity metrics DeepSim and the one proposed by Cheng et al. Appendix A discusses more details of these previous metrics.

__4. Application to more challenging multimodal data is missing__

_Section 3.2_ shows the results of SiamRegQC vs other conventional multimodal similarity metrics Iike
MIND, NCC and deep similarity metrics, DeepSim, and by Cheng for MRI T1-T2 multimodal data.

__5. Method description is short/ambiguous__

Appendix A details a literature survey on the previous deep similarity metrics evaluated for comparison. It also details the architecture and training methodology for SiamRegQC.

__6. Contrastive loss is not elucidated__

Equations 2,3 are expanded to show the exact formulation of the contrastive loss. Section 2 is updated to show the importance of contrastive loss with Figure 2 showing ablated versions of SiamRegQC.

__7. The meaning of ground truth in _Table 2_ is not clear__

_Appendix C and D_ detail the process of sectionwise registration for histology volume reconstruction. We use SiamRegQC to evaluate these sectionwise registrations. The successive sections from the original volume are called ground truth in _Table 2_. More details are added in Appendix C.

__8. The notion of critically evaluating SITK is ambiguous__

_Appendix C_ is revised to explain that SITK is a nontrivial registration algorithm that largely performs well except for some misregistrations as in _Figure 8_. Here SiamRegQC critically evaluates SITK with a lower p-value (a measure of lower confidence in the goodness of registration), whereas MSE and SSIM show high confidence.

Please find these changes highlighted in violet, which incorporate almost all the comments. We are open to any further discussions / clarifications .

---

### Author Response · Authors · 2024-03-18
**Restructuring manuscript to highlight multimodal registration with SiamRegQC as a similarity metric**

Dear Reviewers,

Keeping in mind that our new experiment of multimodal deformable registration with SiamRegQC as a similarity metric is a contribution, we have swapped the last Section of our experiments to accommodate for the multimodal registration contribution. We kindly request you to follow the updated order of the appendix, which will fit well with the general flow of the paper. The revised text is highlighted in a violet font in the manuscript.

Updated Appendix Titles

A- Previous related works

B- Rationale for the architecture of SiamRegQC

C- Continued desirable aspects of SiamRegQC

D- Section-wise Registration evaluation for Nissl-stained volume reconstruction

E- SiamRegQC as a deep similarity metric for VoxelMorph registration of unimodal images.

All the experiments remain the same, and we have updated the reference figures and Appendix in our previous comments (the revised references are in italics).

---

### Author Response · Authors · 2024-03-23

Dear Reviewers,

Thank you for all your reviews. We have addressed all the comments highlighted by you and essential experiments are now part of the main manuscript without deviating from the objectives. We are open to interactions so that there is room for further improvements, given the limited amount of time. We kindly request your consideration in checking if the updated manuscript meets your expectations.

---

### Author Response · Authors · 2024-03-26

Dear reviewers,

We thank all the reviewers for their helpful reviews and suggestions, which are invaluable for elevating the quality of our paper. We would like to kindly remind the reviewers that the discussion period deadline is approaching. We are open to further discussion if there are any more follow-up questions. We sincerely hope for fruitful discussions with all the reviewers, ensuring our responses convincingly address their concerns. We have tried our best to address the reviewers' concerns and remain available to clarify any lingering ambiguities or provide additional information.
Thank you once again for your contributions to our manuscript review process.

---

### Meta-Review · Area_Chair_hiSm · 2024-04-04

**Recommendation:** Accept (Poster)
**Confidence:** 5

**Metareview:**

Although reviewer critiques were significant in their initial ratings (due to the limitation of similarity metrics to rigid registration), the authors significantly improved the paper during the discussion period (extended all experiments to nonlinear registration cases) and responded point-to-point to reviewers' questions. The paper topic should be interesting to MIDL attendees and the paper is in good shape to be discussed at the conference.

---

### Decision · Program_Chairs · 2024-04-05

Accept (Poster)